# Wound Healing Potential of an Oleoresin Essential Oil Chemotype from *Canarium schweinfurthii* Engl.

**DOI:** 10.3390/molecules27227966

**Published:** 2022-11-17

**Authors:** Michel Bonnard, Enzo Martin, Isabelle Parrot

**Affiliations:** 1IBMM, University of Montpellier, CNRS, ENSCM, 1919 Route de Mende, 34095 Montpellier, France; 2FLORE SCOLA, 34980 Saint-Gély-du-Fesc, France

**Keywords:** *Canarium schweinfurthii*, oleoresin, essential oil, terpinolene, *α*-phellandrene, wound healing

## Abstract

This study was conducted to investigate the chemical composition of essential oil (EO) extracted from an oleoresin of *Canarium schweinfurthii* widespread in the Gabonese tropical forest. A great variability in the chemical composition of EO was observed, among which a chemical profile rich in terpinolene and *α*-phellandrene (31.2 and 21.8%, respectively), was found and tested as a natural active ingredient for topical applications. After the evaluation of eye and skin irritancy and sensitization potentials of EO on in vitro and in chemico models, the in vitro modulating potential on a model of wound re-epithelialization was assessed. The terpinolene and *α*-phellandrene-rich chemotype have been proven to accelerate wound healing in a dose-dependent manner (concentration range from 1.8 to 9.0 μg/mL). In addition, the ability of this EO to modulate the pro-inflammatory response in human keratinocytes stimulated by UVB was observed in vitro by the reduction in levels of interleukin 6 (IL-6) and tumour necrosis factor-alpha (TNF-α), suggesting a possible implication during the inflammation phase of wound healing. Despite the high variability in EO composition, a method of solid-phase microextraction (SPME) of the oleoresin headspace is proposed for the in situ identification of the terpinolene and *α*-phellandrene-rich chemotype instead of conducting hydrodistillation. These results offer interesting perspectives for the development of innovative natural ingredients for the topical route, ingredients obtained in an eco-responsible and non-destructive way.

## 1. Introduction

The African elemi *Canarium schweinfurthii* Engl. (Burseraceae) is a massive tree native to tropical Africa, spread over tropical forests of Angola, Cameroon, Congo, Ethiopia, Gabon, Senegal and Tanzania, for example [1,2]. Its fruit, bark and oleoresin are important resources in traditional food, ancestral and medicinal practices. Decoctions, infusions and other herbal mono-preparations or mixtures of leaves, seeds, fruits, bark and oleoresin have been used in folk medicine for the treatment of various diseases such as lung illness, hypertension and ovarian disorder [2,3]. Pharmacological studies conducted on essential oils and plant material extracts have revealed potential antidiabetic [4], antimicrobial [5,6,7,8,9,10], antioxidant [9,11,12,13] and anti-cancer [14,15] activities, in line with the numerous identifications of secondary metabolites reported in the literature (i.e., terpenoids, steroids, glycosides, alkaloids, flavonoids, anthocyanins and tannins) [2,14,15]. In addition, the chemical investigations of extracts from seeds, oleoresins or stem bark have led to isolating new groups of active phytoconstituents such as phenolic and triterpenoid compounds [16,17,18,19]. Given the many possible uses of this tree, its exploitation is highly supervised by local authorities, especially in Gabon, by limiting its overexploitation and thus the risk of deforestation. In response to regulations and local economic needs, the deployment of sustainable harnessing solutions is imperative. In this respect, being non-destructive, the collection or exploitation of the oleoresin to produce essential oil appears to be an excellent option, which offers many new opportunities. Compared to the well-known Asian elemi (*Canarium luzonicum*), the essential oil (EO) extracted from the oleoresin of *Canarium schweinfurthii*, the African elemi (or Aiele), whose yield varies from 4.4 to 7.2% [10,13,20], is almost not commercially available despite its potential uses in aromatherapy or pharmacology [15]. In terms of chemical profile, the EO is described to be mainly composed of monoterpenes and oxygenated compounds and subject to low stability under unscrupulous storage conditions [20,21,22]. Especially, its chemical composition is reported in the literature to be remarkably variable [2], even for EO obtained from oleoresin collected from different regions of a given country [13], which is an evident limitation to its exploitation. Defining the chemical profile and related toxicity of EO is one of the crucial steps before considering any commercial exploitation, especially in fragrance and cosmetic fields where EO harnessing is highly valued.

In this context, we are interested in this study on the chemical composition of an EO obtained from an oleoresin of *C. schweinfurthii* collected in Gabon (Ntoum region), with unprecedented high extraction yield in comparison to other *Canarium* spp. oleoresins described in the literature. The chemical composition of volatiles of the crude oleoresin was determined by headspace solid-phase microextraction combined with gas chromatography and mass spectrometry (HS-SPME-GC-MS) and compared to that of the EO. Since both compositions are not in agreement with those described so far in the literature, showing, in particular, a high content of terpinolene and *α*-phellandrene, both compositions proved to be in agreement with each other. This report highlights a great variability in the chemical composition of the essential oil extracted from oleoresins collected from different source trees located in the same region, and most certainly the presence in Africa of many chemotypes. This study suggests that HS-SPME-GC-MS can be an efficient tool for screening the variability of the extractable EO or highlighting a particular chemotypicity of interest, by simply determining the volatile profile directly from the oleoresin instead of performing a systematic and fastidious extraction. In line with the use of EO as an ingredient in fragrance and cosmetic fields, the skin and eye toxicities of the extracted EO were also investigated in vitro according to the Organization for Economic Co-operation and Development (OECD) reference methods. In addition, the modulating effects of the EO on wound healing were tested in vitro on a model of re-epithelialization. The anti-inflammatory potential was also assessed by the measurement of interleukin 6 and tumour necrosis factor alpha produced by UVB-activated human epidermal keratinocytes and treated with EO. This study establishes for the first time EO extracted from the oleoresin of the Gabonese *C. schweinfurthii* as a natural resource of great potential for applications related to olfaction and wound healing.

## 2. Results

### 2.1. Composition of Volatile Compounds in EO

In this study, we first investigated the production and physicochemical properties of an essential oil obtained from an oleoresin collected from *C. schweinfurthii* in the Ntoum region of Gabon (tree #1). The yellowish EO was produced with a higher yield (12.2%) than those obtained from oleoresins of *Canarium* spp. reported in the literature (up to 7.2%) [10,13,20]. The flash point (46.8 ± 0.3 °C), density (0.8554 ± 0.0001 g/cm^3^ at 20 °C) and refractive index (1.4783 ± 0.0003 at 20 °C) of the EO were also determined. In terms of chemical composition, all 20 compounds detected by GC-MS were identified, revealing a relatively simple composition compared to that of essential oils in general. The retention indices (RI) and the content of identified compounds are listed in Table 1.

Cyclic monoterpene, the major group of compounds identified in EO, was represented by height major compounds (Figure 1), chemically devoid of a functional group: terpinolene (no. 16, 31.23%), *α*-phellandrene (no. 7, 21.88%), limonene (no. 11, 9.09%), sabinene (no. 4, 7.83%), *α*-pinene (no. 2, 5.87%), *β*-phellandrene (no. 12, 5.31%), *p*-cymene (no. 10, 5.28%) and *β*-pinene (no. 5, 3.77%).

It is interesting here to note that the chemical profile of EO obtained in this study is different from those reported in the literature, where a variability is systematically described regardless of the country of origin of the oleoresin [2]. We can mention for example, EO obtained from oleoresins collected in Ugandan, in which the major compounds are *α*-phellandrene (17.9%), *β*-phellandrene (12.9%), *p*-cymene (8.5%), *α*-pinene (4.9%), sabinene (2.9%) and *β*-pinene (2.3%) [22]. Drastically different, the EO obtained from the oleoresin collected in the equatorial rain forest of Central African Republic is rich in oxygenated compounds, composed of octyl acetate (60.0%), (E)-nerolidol (14.0%) and *n*-octanol (9.50%) as the major constituents, molecules which were not detected in our study [20]. This assessment is much pronounced in comparison with oleoresins collected from different regions of Cameroon [21]. As an example, *p*-cymene constituted 9.8% of the EO obtained from the oleoresin collected in Lolodorf, and 25.3% of the EO obtained from the oleoresin collected in Mbouda. In regard to the chemical composition reported for other EO obtained from oleoresins collected in Gabon [24], the distribution of the monoterpenes composition is also significantly different, with limonene (52.1%), sabinene (19.2%), *α*-pinene (10.6%), *p*-cymene (4.3%), *trans*-thuyan-4-ol (3.5%) and *β*-pinene (2.8%) as the major constituents. Moreover, *α*-phellandrene and *β*-phellandrene were detected in low content, whereas they respectively represent 21.8% and 5.3% of the EO studied here. 

Such variability in EO composition was also observed upon analysis of EO obtained from oleoresins collected from the three additional source trees, all located in the Ntoum region (trees #2-#4 in Appendix A). These results, along with data from the literature, demonstrate an important variability in the composition of essential oils obtained from oleoresins of *C. schweinfurthii* depending on the collection area and source tree. A chemical polymorphism within the *schweinfurthii* species seems to emerge. Such a lack of stable volatile profile severely limits the exploitation of oleoresin of *C. schweinfurthii* without a detailed mapping and characterization of potential sources.

### 2.2. Volatile Compounds Identified in the Crude Oleoresin Headspace

With the objective of simplifying chemotype determination and therefore the collection of terpinolene-dominant species, we decided to undertake the direct analysis of headspace volatiles directly on the oleoresin. The volatile compounds of the oleoresin headspace were extracted using solid-phase microextraction and then analyzed by GC-MS. The retention indices (RI) and content of identified compounds are listed in Table 2. A total of 20 compounds were also identified, corresponding to the entire set of compounds detected by GC-MS. As previously observed for the GC-MS analysis of the extracted EO, monoterpene represented the major group of compounds identified in the headspace of the crude oleoresin. Subclasses were in agreement with those previously observed in the corresponding EO. Major compounds were terpinolene (no. 16, 27.86%), *α*-phellandrene (no. 7, 23.29%), sabinene (no. 4, 12.44%), limonene (no. 11, 11.72), *p*-cymene (no. 10, 7.15%), *α*-pinene (no. 2, 5.68%), *β*-pinene (no. 5, 3.95%) and *β*-phellandrene (no. 12, 2.76%). Given the effectiveness of SPME in extracting the set of volatile compounds from the crude oleoresin headspace, employing this strategy would allow the collection of small amounts of oleoresin samples (a few grams) instead of larger quantities required for hydrodistillation, and overall, a more consistent man-to-machine time. Thus, harvesting and screening processes to access a particular chemotype could be clearly simplified.

### 2.3. In Vitro Assessment of Eye and Skin Irritancy

To go further in the potential use of this fragrant essential oil, we were interested in its potential toxicity, especially eye and skin irritation by in vitro evaluation. All validation tests were recorded to conform to the OECD guidelines no. 492 (reconstructed human Cornea-like Epithelium (RhCE) test method for identifying chemicals not requiring classification and labelling for eye irritation or serious eye damage) [25]. A market cytotoxic effect (87% mortality) was recorded after the application of the produced EO on the 3-dimensional reconstituted human corneal epithelial model (Figure 2). In this case, the EO requires a classification category for “eye irritation or serious eye damage” since the viability of tissues was recorded below 60%. Nevertheless, no prediction could be made for category 1 or category 2.

Regarding the evaluation of the in vitro assessment of skin irritancy, all validation tests were recorded to conform to the OECD guidelines no. 439 (in vitro skin irritation: reconstructed human epidermis test method) [26]. No significant cytotoxicity (mortality lower than 10%) was recorded after the application of EO on a 3-dimensional reconstituted human epidermis model (Figure 3). Accordingly, the EO is identified as not requiring classification for skin irritation (UN GHS No Category) since the mean percent tissue viability was higher than 50%.

### 2.4. In Chemico and In Vitro Assessment of Skin Sensitization

In order to pursue the potential use in fragrance or cosmetic fields of the EO obtained from the oleoresin of *Canarium schweinfurthii*, in chemico and in vitro assessment of skin sensitization were undertaken. All validation tests of the in chemico assessment of EO skin sensitization were recorded to conform to the OECD guidelines no. 442C (in chemico skin sensitization: direct peptide reactivity assay) [27]. In the experimental conditions previously described, the cysteine and lysine peptide depletion values were respectively found at 51.5 and 9.9% (raw data available in Appendix A). These results allow for the qualification of the peptide reactivity of EO as positive since its percent peptide depletion is higher than 10% for cysteine and higher than 20% for lysine.

In order to qualify the skin sensitization potential of the produced EO, in vitro assessment was also determined on human leukemia cell line THP-1. All validation tests were also recorded to conform to the OECD guidelines no. 442E (in vitro skin sensitization assays addressing the key event on activation of dendritic cells on the adverse outcome pathway for skin sensitization) [28]. No significant cytotoxicity (cell death ≤ 10%) was observed after the application of EO concentrations ≤ 0.01 mg/mL (raw data available in Appendix A). From 0.05 mg/mL, a does-dependent toxicity was observed. The CV75 value (C_5_ = 0.075 mg/mL) has been calculated using the probit-log[concentration] regression model:Probit(%death) = 1.82 × log[conc.] + 6.38 (r = 0.964)(1)

CD86/CD54 was expressed as follows. The O.D values obtained with each replicate (O.D_meas._) were corrected by subtracting the mean O.D value of NSB blank (O.D_blank_ = 0.114, *n* = 3 for CD86 and O.D_blank_ = 0.116, *n* = 3 for CD54). The corrected O.D values (O.D_cor._) obtained with each replicate in 96-well microplate were multiplied by the ratio of the volume of replicants (400 µL) in 48-well microplate on the volume of replicates (100 µL) in 96-well microplate to obtain O.D_CD86/54_ values per well of 48-well plate. The mean and standard deviation of [O.D_cor./well_] values were calculated. The fluorescence intensities [FI_mes_] recorded with each replicate in Resazurin plate were corrected by subtracting the mean FI value of blank (FI_Blank_ = 2967, *n* = 3). The mean fluorescence intensity was calculated: [FIM_cor_] (*n* = 3). Results ([CD86 (%)] and [CD54 (%)] mean values (n = 3) obtained in a single experiment are summarized in Table 3. The [CD86 (%)] values obtained (≥150 with the 6 tested concentrations) and the [CD54 (%)] values obtained (≥200 with the 6 tested concentrations) allowed to qualify the dendritic cell activation potential of the EO as positive. 

Altogether, the in chemico and in vitro skin sensitization assays show that the EO extracted from the crude oleoresin of *C. schweinfurthii* can activate the human leukemia cell line THP-1 used as a substitute for cutaneous dendritic cell. Accordingly, the combination of the two assays allows the qualification of the EO as an in vitro skin sensitizer. 

### 2.5. Cytotoxicity on HaCaT Keratinocytes

If the qualification as a skin sensitizer was probably expected considering the presence of limonene, but also the fact that EO is a concentrated ingredient, we were then interested in the determination of its cytotoxicity in order to further investigate the skin wound healing and anti-inflammatory potential. The cytotoxic determination of EO on HaCaT keratinocytes, evaluated from the broad concentration range for an incubation time of 24 h, revealed no cytotoxic effect with EO concentrations ≤ 156.25 µg/mL (Table 4). From 312.5 µg/mL, EO has a dose-dependent cytotoxic effect. Regarding the cytotoxic determination of EO conduced from the narrow range (48 h incubation time), results show no significant cytotoxic effect with concentration ≤ 7.5 µg/mL (Table 5). From 15 µg/mL, EO has a dose-dependent cytotoxic effect.

The concentration of EO showing 90% cell viability was calculated from the dose-response curve using linear probit-log regression model after conversion of % cell death into probit and by plotting probit value against logarithm of concentration. A EO concentration of 9.14 µg/mL was obtained. The EO concentration of 9 µg/mL was fixed as the maximal assay dose for both the scratch wound healing assay and the anti-inflammatory potential evaluation.

### 2.6. In Vitro Assessment of the Modulating Effects on Skin Wound Healing

The two major compounds identified in EO (terpinolene and *α*-phellandrene), constituting more than 50% of EO, being described in the literature to increase the migration and proliferation of fibroblasts in vitro [29], suggest promising wound healing properties. Therefore, we set out to test the modulating effect of EO on human epidermal keratinocytes (HaCaT cells) using a scratch assay. The HaCaT cells were exposed to three EO concentrations (1.8, 4.5 and 9.0 µg/mL) immediately after wound scratching, i.e., at time zero (T_0_). The area of each wound was measured daily by placing culture dish under video analysis system at D_0_ (T_0_ and T_1_), D_1_ (T_2_ and T_3_) and D_2_ (T_4_). Results (denuded area) are expressed in arbitrary units. The percentage of the initial area still denuded was calculated for each selected field and averaged for each recording time (Table 6).

A progressive reduction in wound area was observed in the EtOH control group. It is noteworthy that the complete closure of the denuded area was observed after 2 days of incubation. At T_3_, the mean wound area represented 35% of the initial area. The kinetic wound healing recorded with C_EO-1_, C_EO-2_ and C_EO-3_ was faster than that observed with both the negative control and the EtOH control groups (Figure 4). The inducing effect was dose dependent in the range of the tested concentrations. The differences recorded at the T_2_ and T_3_ recoding time in C_EO-1_, C_EO-2_ and C_EO-3_ groups were statistically significant (*p* ≤ 0.01, Student test) comparatively to the EtOH control. A complete closure of the denuded area was observed at T_3_ for C_EO-3_ (9.0 µg/mL).

The analysis of data indicated that the treatment of HaCaT cells with EO in the range of the tested concentrations (from 1.8 to 9.0 µg/mL), accelerated very significantly the time-course of re-epithelialization at T_1_, T_2_ and T_3_ compared to the negative and EtOH control groups. In the same experimental condition, the kinetics of wound healing recorded with HB-EGF (100 ng/mL) was faster than that observed with the negative and EtOH control groups (Table 6, Figure 4). The differences recorded at T_1_, T_2_ and T_3_ proved to be statistically significant (*p* ≤ 0.01, Student test) comparatively to the negative and EtOH control groups. Data also indicated that treatment of HaCaT keratinocytes with EO in the tested concentration range led to a higher re-epithelialization at 32 h compared to treatment with HB-EGF at 100 ng/mL. In addition, it is noteworthy that the kinetics of wound healing recorded in both negative and EtOH control groups were almost identical (no statistical difference excepted at T_3_). These results show the promising wound healing potential of EO, possibly attributable to its high terpinolene and *α*-phellandrene content, two compounds known to favor wound healing, as reported in the literature [29].

### 2.7. In Vitro Assessment of Anti-Inflammatory Potential on the UV-Induced Inflammatory Response

The wound healing process can be divided into three interdependent phases, including inflammation [30]. In order to evaluate the involvement of EO in the underlying process of wound healing, its effect was tested on the expression of the pro-inflammatory cytokines IL-6 and TNF-α. This assay was performed on human epidermal keratinocytes exposed to three EO nontoxic concentrations (1.8, 4.5 and 9.0 µg/mL). The IL-6 and TNF-α levels were quantified 24 h after UVB-irradiation. IL-6 levels (pg/well) were interpolated from the standard curve built with IL-6 standard provided with the Elisa kit. Mean of IL-6 levels (pg/well) was calculated in each experimental group. The total protein level (µg/well) was interpolated from the standard curve built with the BSA standard. The mean of total protein levels was calculated in each experimental group. The mean of IL-6 levels was standardized to the mean of protein contents. The final results were expressed in pg of IL-6 per mg of protein (Table 7). The UV-induced IL-6 level was calculated as:IL-6 induction = IL-6_(+)UV_ − IL-6_(−)UV_(2)

Regarding the control (+)UV, UVB induced an important increase in IL-6 release. Baseline cytokine level was increased by 21.6 times after stimulation by UVB. From the HaCaT cells treated with EO, a statistically significant decrease in IL-6 level was observed in C_EO-1_, C_EO-2_ and C_EO-3_ (*p* ≤ 0.01, Student test), comparatively to the control (+)UV. The down-regulating effect was not dose dependent, the tested concentrations appeared quite equivalent: C_EO-1_ (−56%), C_EO-2_ and C_EO-3_ (−51%). In the same experimental conditions, dexamethasone (100 µM) inhibited by 92% of the UV-induced IL-6, thus validating the assay.

The levels of TNF-α (pg/well) were interpolated from the standard curve built with TNF-α standard provided with the Elisa kit. The mean of TNF-α levels (pg/well) was calculated in each experimental group. The total protein level (µg/well) were interpolated from the standard curve built with the BSA standard. The mean of total protein levels was calculated in each experimental group. The mean of TNF-α levels were standardized to the mean of protein of protein contents. The final results were expressed in pg of TNF-α per mg of protein (Table 8). The UV-induced TNF-α level was calculated as:TNF-α induction = TNF-α_(+)UV_ − TNF-α_(−)UV_(3)

Regarding the control (+)UV, UVB induced a marked increase in TNF-α release. Baseline level was increased by 10.6 times after UVB stimulation. From the HaCaT cells treated with EO, the recorded levels of TNF-α were lower than that recorded in the control (+)UV. The inhibitor effect was inversely proportional to the dose applied: C_EO-1_ (−36%), C_EO-2_ (−25%) and C_EO-3_ (−14%). The differences observed in C_EO-1_, C_EO-2_ and C_EO-3_ groups were statistically significant (*p* ≤ 0.01, Student test), comparatively to the control (+)UV. In the same experimental conditions, dexamethasone (100 µM) was inhibited by 53% of the UV-induced TNF-α. In this study, the production of IL-6 and TNF-α by UVB-stimulated human epidermal keratinocytes was significantly reduced by EO, suggesting a potential involvement in wound healing by reducing or suppressing the overproduction of these pro-inflammatory cytokines.

## 3. Discussion

As illustrated by data reported in the literature, the composition of essential oils obtained from the oleoresin of *Canarium schweinfurthii* appears highly variable. Indeed, while the EO obtained from an oleoresin collected in Cameroon is mainly composed of limonene and *α*-terpineol as major compounds (42.7 and 34.4%, respectively) [13], that collected in Central African Republic was described as rich in octyl acetate and (E)-nerolidol (60.0 and 14.0%, respectively) [20]. The composition of the EO studied here is additional evidence of this important variability. Such variation can arise from multiple factors. If the local environment of *C. schweinfurthii* trees (humidity, temperature, sunlight, predation, surrounding biodiversity, …) can obviously lead to important variation, the vegetative stage, the frequency and the seasonality of resin collection could also contribute to such variability and deserve to be further investigated. Another factor never mentioned in phytochemical studies of *C. schweinfurthii* could also emerge: the sexual type, with the female type only capable of fructification (not systematic) [31]. 

The study developed here highlighted a very specific chemotype and also an interesting biological activity for topical applications to be considered. The wound healing potential of the terpinolene- and *α*–phellandrene-rich EO chemotype described here, can be potentially related to a specific implication of these two biomolecules during the inflammation phase, especially due to their well-known reducing effect on IL-6 and TNF-*α* levels, already reported in the literature [29]. If the anti-inflammatory potential of EO obtained from an oleoresin collected in Cameroon was already mentioned in the literature [13], the low content of terpinolene and *α*–phellandrene and its inhibition effect tested on 5-lipoxygenase cannot allow establishing a direct connection between the two studies. The lack of stable and easily identifiable chemotypes resins/trees of *C. schweinfurthii* appears as a limiting factor for the exploitation of oleoresin EO from *C. schweinfurthii*. However, the use of a HS-SPME-GC-MS method developed in our study, which can be directly conducted on oleoresins, allows considering a systematization of the identification of EO chemotypes of interest for specific activities previously described in the literature, such as antiradical, anti-inflammatory, antioxidant, antimicrobial [11,13], analgesic [20], antifungal [21] and anti-termitic [22].

## 4. Materials and Methods

### 4.1. Materials

All chemicals and reagents were used without any further purification. Anhydrous sodium sulfate (99% purity) was purchased from VWR Chemicals (Radnor, PA, USA). Argon (5.0 purity/99.999%) was purchased from Linde Gas (Saint-Priest, France). Helium used for GC analyses (6.0 purity/99.9999%) was purchased from Linde Gas (Saint-Priest, France). C_7_–C_30_ saturated *n*-alkanes analytical standard (1000 µg/mL of each component in hexane) was purchased from Sigma-Aldrich (Saint-Quentin-Fallavier, France). Butylated hydroxytoluene (BHT), limonene, α-pinene, α-terpineol and *p*-cymene were purchased from Sigma-Aldrich (Saint-Quentin-Fallavier, France). *β*-pinene and *γ*-terpinene were purchased from Honeywell Fluka™ (Charlotte, NC, USA). *δ*^3^-carene and eucalyptol were purchased from PCW (Grasse, France).

### 4.2. Collection of the Crude Oleoresin

The crude oleoresin samples used in this study were selected from source trees in the region of Ntoum (Gabonese tropical forest, trees #1 to #4) and were supplied by Pr Jean-Noël Gassita (professor of pharmacognosy and pharmacology, university of Libreville, curator of the Arboretum of Sibang) with permission of the Ministry of Environment of Gabon (permission number 000720). The fresh exuding oleoresins were collected by hand from wounds left by recent carving (October–November 2018). The oleoresin sample batches (~2 kg each) were immediately stored in cling film, transported to the laboratory by plane and kept in the dark at 4 °C until EO extraction in order to minimize the degradation and loss of volatile compounds.

### 4.3. Essential Oil Extraction

The crude oleoresin (360 g) was roughly crushed by hand prior to EO extraction. The EO was obtained by hydrodistillation using an all-glass 5 L Clevenger-type apparatus. After complete extraction (~4 h), the EO was separated from the aqueous phase, dried with anhydrous sodium sulfate and filtered on cotton wool. The essential oil was kept under argon for stabilization and finally stored in the dark at 4 °C until analysis. The extraction yield (%, *w*/*w*) was based on the crude oleoresin weight. The flash point of EO was measured with a Setaflash™ Series 3 (Stanhope-Seta, Chertsey, UK), the density was obtained with a Densito™ 30PX (Mettler Toledo, Columbus, OH, USA) and the refractive index was determined with an Abbe refractometer (Paralux, Geispolsheim, France). Each physicochemical determination was carried out in triplicate.

### 4.4. Identification of Volatile Compounds in EO by GC-MS

The identification of volatile compounds in EO (tree #1) was conducted in both polar and non-polar stationary phases. For identification using non-polar stationary phase, EO was analyzed with a TRACE 1300 gas chromatography system (Thermo Fisher Scientific, Waltham, MA, USA) equipped with a 30 m × 0.250 mm i.d. × 0.25 µm film thickness, DB-5MS column (Agilent Technologies, Santa Clara, CA, USA). For EO injection (1 µL), GC was run in split mode with a 1/100 ratio. The injector temperature was set at 250 °C and the oven temperature was programmed from 70 °C (hold for 3 min) to 200 °C at 7 °C/min (with a final hold time of 5 min). Helium was used as the carrier gas at a constant flow rate of 0.9 mL/min. Ionization and detection were achieved with a DSQ II single quadrupole mass spectrometer (Thermo Fisher Scientific, Waltham, MA, USA) in electron impact ionization mode (70 eV). MS data were recorded in full scan mode (*m*/*z* range from 33 to 450). Transfer line and MS ion source temperatures were set at 280 °C and 250 °C, respectively. The identification of volatiles in EO using a polar stationary phase was conducted with an Agilent 7820A gas chromatography system (Agilent Technologies, Palo Alto, CA, USA) equipped with a 20 m × 0.180 mm i.d. × 0.18 µm film thickness, HP-Innowax column (Agilent Technologies, Santa Clara, CA, USA). For EO injection (1 µL), GC was run in split mode with a 1/20 ratio. The injector temperature was set at 250 °C and the oven temperature was programmed from 70 °C (hold for 2 min) to 250 °C at 10 °C/min (with a final hold time of 5 min). Helium was used as the carrier gas at a constant flow rate of 0.9 mL/min. Ionization and detection were achieved with a 5977E MSD single quadrupole mass spectrometer (Agilent Technologies, Palo Alto, CA, USA) in electron impact ionization mode (70 eV). MS data were recorded in full scan mode (*m*/*z* range from 33 to 350). Transfer line and MS ion source temperatures were set at 280 °C and 230 °C, respectively. The Kováts retention index (RI) of detected compounds was calculated using a C_7_–C_30_ saturated *n*-alkanes analytical standard. Compounds identification was achieved by comparison of their mass spectra and RI to those reported in a mass spectral library developed at the laboratory, with a GC-MS data derived from the analysis of commercial essential oils and with the National Institute of Standards and Technology (NIST) online database. Experiments were performed in triplicate.

### 4.5. Quantification of Volatile Compounds in EO by GC-MS

The compounds identified in EO (tree #1) were quantified according to the internal standard method using the non-polar stationary phase method and with BHT as the internal standard. Molecules with similar chemical structures were gathered and quantified, with one pure analytical standard supposing an equal response factor in each molecules group. All compounds were distributed as follows: bicyclic monoterpenes with an intracyclic alkene function (*α*-thujene, *α*-pinene and *δ*^3^-carene); bicyclic monoterpenes with an extracyclic alkene function (camphene, sabinene and *β*-pinene); cyclic monoterpenes with at least one intracyclic alkene function (2-menthene, *α*-phellandrene, *α*-terpinene and *γ*-terpinene); aromatic compounds (*p*-cymene and *p*-cymenene); cyclic monoterpenes with two alkene functions, one intracyclic and one extracyclic (limonene, *β*-phellandrene and terpinolene); bicyclic ethers (eucalyptol); bicyclic or cyclic alcohols (4-thujanol, terpinen-4-ol and *α*-terpineol). Standard solutions containing all the analytical references (*α*-pinene, *β*-pinene, *γ*-terpinene, *δ*^3^-carene, *p*-cymene, limonene, eucalyptol and *α*-terpineol) were prepared at 6 different concentrations (from 0.5 to 500 ppm in each standard) to obtain the calibration curves. Two different EO concentrations (500 mg/L for major and 5000 mg/L for minor compounds) were used. Three controls were injected before and after EO in order to confirm the calibration curve. Each solution (containing 400 ppm of the internal standard) was analyzed in triplicate and results are expressed as mean ± standard deviation. For quantification, MS data were recorded in selected ion monitoring (SIM) mode (*m*/*z* at 93, 117, 154 and 205). These *m*/*z* were used to quantify the different EO compounds depending on their mass spectra, as detailed in Table 1.

### 4.6. Identification of Volatile Compounds from the Crude Oleoresin by HS-SPME-GC-MS

SPME was performed using a manual system equipped with a 50/30 µm DVB/CAR/PDMS fiber (Supelco Inc., Bellefonte, PA, USA). The crude oleoresin (approximately 0.8 g, tree #1) was placed into a 40 mL vial (Phenomenex Inc., Torrance, CA, USA) sealed with a PTFE/silicone septum (Thermo Fisher Scientific, Waltham, MA, USA). After 5 min of equilibration, the fiber was slotted for 5 min for volatile compounds extraction and inserted through a septum at the GC manual injection port for 5 min at 250 °C for complete desorption. For identification using non-polar stationary phase, volatiles of the crude oleoresin were analyzed with a TRACE 1300 gas chromatography system (Thermo Fisher Scientific, Waltham, MA, USA) equipped with a 30 m × 0.250 mm i.d. × 0.25 µm film thickness, DB-5MS column (Agilent Technologies, Santa Clara, CA, USA). GC was run in split mode with a 1/200 ratio. The injector temperature was set at 250 °C and oven temperature was programmed from 50 °C (hold for 5 min) to 250 °C at 10 °C/min (with a final hold time of 5 min). Helium was used as the carrier gas at a constant flow rate of 0.9 mL/min. Ionization and detection were achieved with a DSQ II single quadrupole mass spectrometer (Thermo Fisher Scientific, Waltham, MA, USA) in electron impact ionization mode (70 eV). MS data were recorded in full scan mode (*m*/*z* range from 33 to 350). Transfer line and MS ion source temperatures were set at 280 °C and 250 °C, respectively. For identification using polar stationary phase, volatiles of the crude oleoresin were analyzed with an Agilent 7820A gas chromatography system (Agilent Technologies, Palo Alto, CA, USA) equipped with an Agilent HP-INNOWax column (0.180 mm × 20 m × 0.18 µm). GC was run in split mode with a 1/10 ratio. The injector temperature was set at 250 °C and oven temperature was programmed from 50 °C (hold for 5 min) to 250 °C at 10 °C/min (with a final hold time of 5 min). Helium was used as the carrier gas at a constant flow rate of 0.9 mL/min. Ionization and detection were achieved with a 5977E MSD single quadrupole mass spectrometer (Agilent Technologies, Palo Alto, CA, USA) in electron impact ionization mode (70 eV). MS data were recorded in full scan mode (*m*/*z* range from 33 to 350). Transfer line and MS ion source temperatures were set at 280 °C and 230 °C, respectively. Regarding the RI calculation, a C_7_–C_30_ saturated *n*-alkanes analytical standard was also used. A volume of 50 µL was placed into a 10 mL sealed vial (Phenomenex Inc., Torrance, CA, USA) and heated to 50 °C. After 5 min of equilibration, the fiber was slotted for 15 min for alkanes extraction. Compounds identification was achieved by comparison of their mass spectra and RI to those reported in a mass spectral library developed at the laboratory, with a GC-MS data derived from the analysis of commercial essential oils and with the NIST online database. Analysis was conducted three times and results are expressed as mean ± standard deviation.

### 4.7. In Vitro Assessment of Eye and Skin Irritancy

The eye irritation potential of EO (tree #1) was assessed by the measurement of its cytotoxic effect on a 3-dimensional reconstituted human corneal epithelial model (HCE, SkinEthic^TM^, EPISKIN SA, France), according to the OECD guideline no. 492. Briefly, the pure EO was applied topically to tissues for 30 min. The experimental groups consisted of an EO group (2 HCE tissue replicates treated with EO), a negative control group (2 HCE tissue replicates treated with PBS without Ca^2+^ and Mg^2+^) and a positive control (2 HCE tissue replicates with methyl acetate). The eye irritation potential of EO was predicted by the mean percent tissue viability normalized to the negative control which was set at 100%. The skin irritation potential of EO was assessed by the measurement of its cytotoxic effect on a 3-dimensional reconstituted human epidermis model (EpiSkin^TM^, EPISKIN SA, France), according to the OECD guideline no. 439. Briefly, the pure EO was applied topically to the epidermis units for 15 min. The experimental groups consisted of an EO group (3 tissues treated with EO), a negative control group (3 tissues receiving a buffered saline PBS solution) and a positive control (3 tissues treated with 5% laurylsulfate solution). The skin irritation potential of EO was predicted by the mean percent tissue viability normalized to the negative control which was set at 100%.

### 4.8. In Vitro Assessment of Skin Sensitization

The skin sensitization potential of EO (tree #1) was estimated by quantification of changes in the expression of the cell surface markers, namely CD86 and CD54, associated with the process of activation of dendritic cells in the human leukemia cell line THP1 after exposition to EO at sub-cytotoxic concentrations. The changes in surface marker expression were measured by immune-cytochemical analysis conducted concurrently to cytotoxicity measurement, according to the h-CLAT test (OECD guideline no. 442E). Briefly, six EO solutions were prepared on a volume basis using DMSO, EtOH (C_6_ = 1.2 × C_5_; C_5_ = 1.2^0^ × C_5_ = CV75 or [DNC_THP-1_]; C_4_ = 1.2^−1^ × C_5_; C_3_ = 1.2^−2^ × C_5_; C_2_ = 1.2^−3^ × C_5_; C_1_ = 1.2^−4^ × C_5_; with [DNC_THP−1_]: nontoxic dose in THP1-CV75 assay). Concurrently to the EO assay, one positive control was used in the experiment (2,4-dinitrochlorobenzene at 5 µg/mL). A volume of 100 µL of working solutions of EO was mixed 1:1 (*v*/*v*) with the THP1 cell suspension in a 96-well flat-bottom plate. The treated plate was then incubated for 24 h (±2 h) at 37 °C (±1 °C) under 5% CO_2_. After 24 h of exposure to the test item, 10 µL per well of CellTiter-Blue Reagent (Promega) was added. After shaking, the 96-well plate was incubated at 37 °C (±1 °C) under 5% CO_2_, for 90 min (±5 min) to allow reduction reactions of the detection reagent (resazurin) by metabolically active cells to fluorescent resorufin. The fluorescence produced and monitored at λ_ex_/λ_em_ = 535/590 nm was proportional to the number of viable cells. The fluorescence intensities [FI_mes_] of measured wells were corrected by subtracting the mean [FI_NSB_] value of non-specific fluorescence blank (*n* = 4). The geometric mean fluorescence intensity was calculated: [FIM_cor_] (*n* = 4). The percentage of cell viability (Viab.%) was calculated for each concentration tested, according to:(4)$$\mathrm{Viab}.\%=\frac{{\mathrm{FIM}}_{\mathrm{cor}}\left[\mathrm{Treated}\;\mathrm{wells}\right]}{{\mathrm{FIM}}_{\mathrm{cor}}\left[\mathrm{Control}\;\mathrm{wells}\right]}\times 100$$

The concentration of EO showing 75% of THP1 cell survival (25% cytotoxicity) was calculated from the dose-response curve using a linear probit-log regression model after conversion of the % cell death into probit referring to the probit table and by plotting the probit value against logarithm of the concentration. For each cell marker of interest, the optical densities [O.D_CD86/CD54_] of wells recorded in CD86 and CD54 assays were corrected by subtracting the mean O.D value of NSB (no specific binding) blank (*n* = 3). The geometric mean [O.D_corr._] was calculated in each assay (*n* = 3). The relative expression level of CD86 and CD54 was calculated in each experimental condition, according to:(5)$$\mathrm{CD}86/\mathrm{CD}54\;\left(\%\right)=\frac{O.D_{\mathrm{corr}.}{\mathrm{mean}}^{\left[\mathrm{CD}86/\mathrm{CD}54\;\mathrm{assay}\right]}}{\mathrm{Viab}.\%}\times 100$$

### 4.9. In Chemico Assessment of Skin Sensitization

The EO skin sensitization potential (tree #1) was estimated by measuring the remaining concentration of synthetic cysteine and lysine peptides models (JPT Peptide Technologies) in the reaction medium after being reacted with the EO during 24 h, according to the DPRA test (OECD guideline no. 442C). Briefly, one negative control (phosphate buffer pH 7.5) and three reference controls were included in [Lys-Pep] assay:

[EO control]: wells containing only EO (wo [Lys-Pep])

[peptide control]: wells containing only [Lys-Pep] (wo test item)

[solvent control]: wells containing only the solvents (solvent_[Lys-Pep]_) + (solvent_[EO]_)

One negative control (isopropanol) and three reference controls were included in [Cys-Pep] assay:

[EO control]: wells containing only EO (wo [Cys-Pep]

[peptide control]: wells containing only [Cys-Pep] (wo test item)

[solvent control]: wells containing only the solvents (solvent_[Cys-Pep]_) + (solvent_[EO]_)

Two positive controls were tested in each experiment:

[positive control #1]: p-phenylenediamine (PPD, 2 mM)

[positive control #2]: cinnamaldehyde (CinAld, 2 mM)

Each of the EO, positive and negative control were analyzed in triplicate for both peptides. A standard calibration curve was generated for both peptides. Peptide standards were prepared in phosphate buffer (pH 7.5) for [Cys-Pep] and phosphate buffer (pH 10.0) for [Lys-Pep]. Using serial dilution standards of the peptide stock-solution (400 µM for [Cys-Pep], 200 µM for [Lys-Pep]), 6 calibration solutions (St_1–_–_6_) were prepared to cover the ranges from 12.5 to 400 µM for [Cys-Pep] and from 6.25 to 200 µM for [Lys-Pep]. A blank with the dilution buffer (St_0_) was also included in the standard calibration curves. The peptide reactivity of the test item was expressed as the peptide depletion ratio after incubation of peptide with EO. For each sample, the mean fluorescence intensity (FI) of replicates (*n* = 3) measured in the presence of peptide was corrected by subtracting the mean FI value of replicates (*n* = 3) measured in absence of peptide:FI_corr_ = FI_(+)_ − FI_(−)_(6)

From FI_corr_ values, the concentration of peptide was calculated by extrapolation from the linear calibration curve obtained from peptide standards. The percent peptide depletion (PD%) was calculated for each model peptide, according to:(7)$$\mathrm{PD}\%=\left(1-\frac{{\mathrm{FI}}_{\mathrm{corr}}\mathrm{mean}\left[\mathrm{treated}\right]}{{\mathrm{FI}}_{\mathrm{corr}}\mathrm{mean}\left[\mathrm{control}\right]}\right)*100$$

### 4.10. In Vitro Assessment of the Modulating Effects on Skin Wound Healing

The modulating effects of EO (tree #1) on the skin wound healing process were evaluated by measuring the migration of HaCaT keratinocytes in an artificial wounded cultured keratinocyte monolayer (scratch wound healing assay). The cytotoxicity of EO was first evaluated in order to define the highest tolerated dose at which no reduction in cell viability is observed. Cell viability was assessed by a neutral red uptake test. Neutral red is a vital dye that readily penetrates cell membranes by non-ionic diffusion and then accumulates intracellularly in lysosomes of living cells. The uptake and retention of neutral red are modified when effects directed especially at the lysosomal membrane occur. These changes brought about by the action of xenobiotics result in a decreased uptake and binding of neutral red, allowing distinction between viable and damaged or dead cells. The principle of the neutral red uptake assay consists in incubating cells in neutral red solution. The optical density (O.D) values of the resulting neutral red extract measured by spectrophotometry at 540 nm indicate the number of viable cells after toxic insult. Before confluence, cells were removed with trypsin and suspended in culture medium for counting. The cells were seeded into cell culture medium containing EO. After exposure to increasing concentration of EO, cell viability was determined by the neutral red uptake. The cytotoxicity of EO was assessed in two steps with two different ranges of test concentrations (broad and narrow range). The mean O.D of each set of replicates was determined for each EO concentration and expressed as absorbance observed as % of control cultures. Cell viability was calculated for each concentration of EO. The broad range cytotoxicity was determined on HaCaT keratinocytes detached with trypsin and suspended in a 10% fetal calf serum medium and for an incubation time of 24 h. A 500 mg/mL stock solution of EO was prepared in ethanol (EtOH) and successively diluted in the 10% fetal calf serum medium (9 concentrations from 5000 to 19.53 µg/mL). The narrow range cytotoxicity was determined for an incubation time of 48 h with EO at 18 concentrations varying from 600 to 0.23 µg/mL. Results are expressed as mean ± standard deviation.

The scratch wound healing assay is based on the evaluation of the closure of a scratch wound in a confluent cell monolayer by the migration of cells cultured in the absence or presence of EO. The basic steps of the scratch assay involve the creation of an artificial gap, so called scratch, in a straight line with p200 pipet tip on a 100% confluent cell monolayer, capture of images at the beginning and regular intervals during cell migration to close the scratch and comparison of the images for measuring cell migration. The scratch assay was conducted on HaCaT keratinocytes detached with trypsin and suspended in a 10% fetal calf serum medium and for an incubation time of 72 h. The effects of EO were tested at 3 non-toxic concentrations and exposed to HaCaT cells immediately after wound scratching. Concurrently, an EtOH control and a negative control were also included in the experimental design, as well as a positive control consisting of heparin-binding EGF-like growth factor (HB-EGF) at 100 ng/mL. Experiments were performed in triplicate and results are expressed as mean ± standard deviation and statistical significance was calculated using Student test (*p* ≤ 0.01).

### 4.11. In Vitro Assessment of Anti-Inflammatory Potential on the UV-Induced Inflammatory Response

The anti-inflammatory potential of EO (tree #1) on the inflammatory response in human cutaneous cells was evaluated by the measurement of the production of interleukin 6 (IL-6) and tumour necrosis factor alpha (TNF-α) by UVB-activated human epidermal HaCaT keratinocytes. After the cytotoxic determination previously detailed, HaCaT keratinocytes were detached with trypsin and suspended in a 10% fetal calf serum. UV stress was generated by UVB exposure (20 mJ/cm^2^) using parallel Philips TL 20 W/12 tubes. The EO was tested at 3 non-toxic concentrations with an incubation time of 48 h. Control groups were carried out in parallel:-2 non stressed controls: cells incubated in 10% fetal calf serum and 10% fetal calf serum + 1% EtOH without UV exposure.-1 stressed control: cells incubated in 10% fetal calf serum + 1% EtOH with UV exposure.

At D_0_, 48 h after plating, cells monolayers were washed with PBS and exposed to UVB in the presence of PBS. Immediately after irradiation, PBS was replaced by fresh medium containing EO. Cells were then replaced in the incubator at 37 °C for 24 h in a humidified 5% CO_2_ atmosphere (treatment post-UV). At the end of the assay (D_1_), the conditioned media were collected, aliquoted and stored at −20 °C until analysis of IL-6 and TNF-α by Elisa kit. Concurrently, the corresponding HaCaT monolayers were rinsed with HBSS and extracted to quantitate cellular proteins content. A NaOH solution (0.1 N) was added into each well. After incubation, aliquots of lysates were collected and stored at −20 °C until quantitating proteins content using BCA protein assay kit (Pierce, Thermo Fisher Scientific, Sweden).

Concurrently, dexamethasone (Sigma-Aldrich, Saint-Quentin-Fallavier, France, ref. D1756) was tested at 100 µM as a positive control. Experiments were performed in triplicate. The IL-6 content in incubation media was measured using an Elisa assay kit (IL-6 high sensitivity Elisa kit, Enzo). The absorbance was recorded at 450 nm. The TNF-α content in incubation media was measured using an Elisa assay kit (human TNF-α high sensitivity Elisa kit, R&D Systems, USA MN). The absorbance was recorded at 450 nm. After thawing, aliquots of each well cell extracts were transferred to a 96-well microplate. 200 µL of BCA reagent (BCA protein assay, Pierce, Thermo Fisher Scientific, Sweden) was added. After incubation for 30 min, O.D were measured at 570 nm. Results are expressed as mean ± standard deviation and statistical significance was calculated using Student test (*p* ≤ 0.01).

## 5. Conclusions

In the present study, an essential oil with a terpinolene- and phellandrene-rich chemotype was obtained from the oleoresin of *C. schweinfurthii*. The modulating potential of this EO on wound healing process was investigated in an in vitro model of wound re-epithelialization by migration of human epidermal keratinocytes. Under the experimental conditions described here, this study has shown that the essential oil, in the range of tested concentration (between 1.8 and 9.0 µg/mL), has the capacity to accelerate wound healing significantly and dose-dependently, possibly related to the modulation of levels of pro-inflammatory cytokines. Despite the high variability of EO composition obtained from different source trees, the use of SPME for the direct analysis of the oleoresin headspace, representative of the extractable EO, has revealed to be an efficient screening method for the identification or selection of oleoresin chemotype with wound healing potential. This study provides a new insight for future topical applications and products using a natural ingredient, whose collection is not destructive of the natural resource, without the use of toxic solvent and with high extraction yield.

To better consider the exploitability of the oleoresin, future investigations can be envisaged, especially the study of seasonal oleoresin collections, at different locations in Gabon and/or from female- and male-type trees. In addition, the determination of the phytochemical composition of the oleoresin by-product generated during hydrodistillation may extend the potential applications, especially in the case of identification of bioactive compounds. Moreover, an in vivo evaluation of the wound healing capacity of the terpinolene- and α-phellandrene-rich EO chemotype may definitely establish this chemotype as a wound healing ingredient for topical applications.

## Figures and Tables

**Figure 1 molecules-27-07966-f001:**
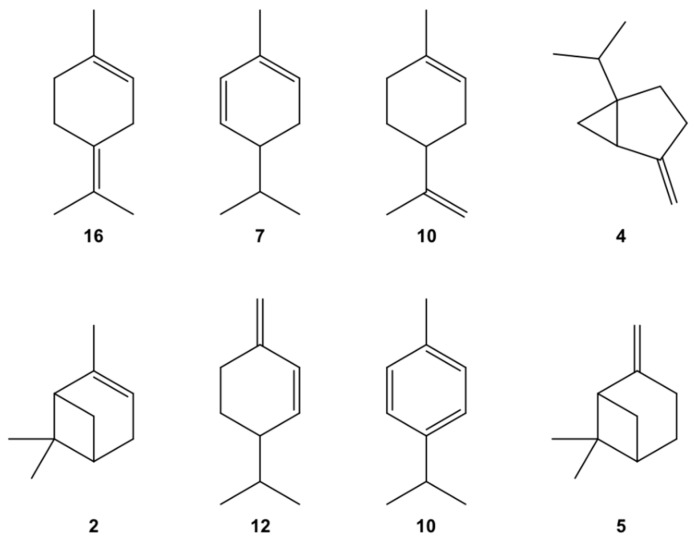
Structure of major compounds identified in an EO obtained from *C. schweinfurthii* oleoresin collected in Gabon: terpinolene (no. 16), *α*-phellandrene (no. 7), limonene (no. 11), sabinene (no. 4), *α*-pinene (no. 2), *β*-phellandrene (no. 12), *p*-cymene (no. 10) and *β*-pinene (no. 5).

**Figure 2 molecules-27-07966-f002:**
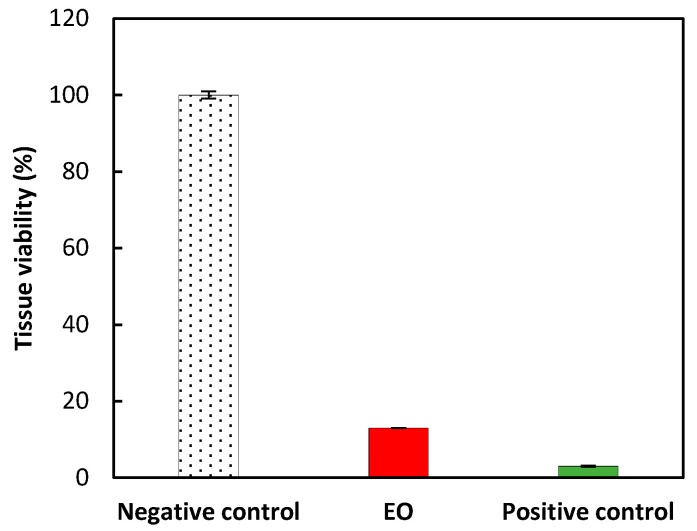
Tissue viability was recorded from the eye irritation test. Negative control (methyl acetate, 100% viability), EO (13% viability) and positive control (PBS without Ca^2+^/Mg^2+^, 3% viability). Results are expressed as mean ± standard deviation (triplicate).

**Figure 3 molecules-27-07966-f003:**
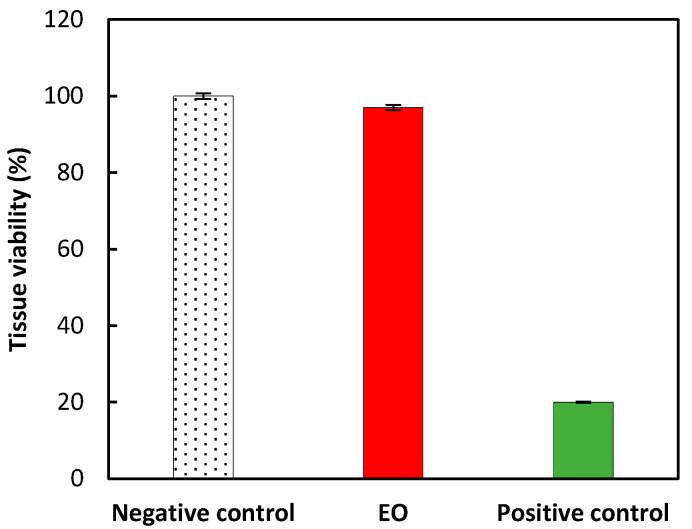
Tissue viability was recorded from the skin irritation test. Negative control (PBS, 100% viability), EO (97% viability) and positive control (5% SDS, 20% viability). Results are expressed as mean ± standard deviation (triplicate).

**Figure 4 molecules-27-07966-f004:**
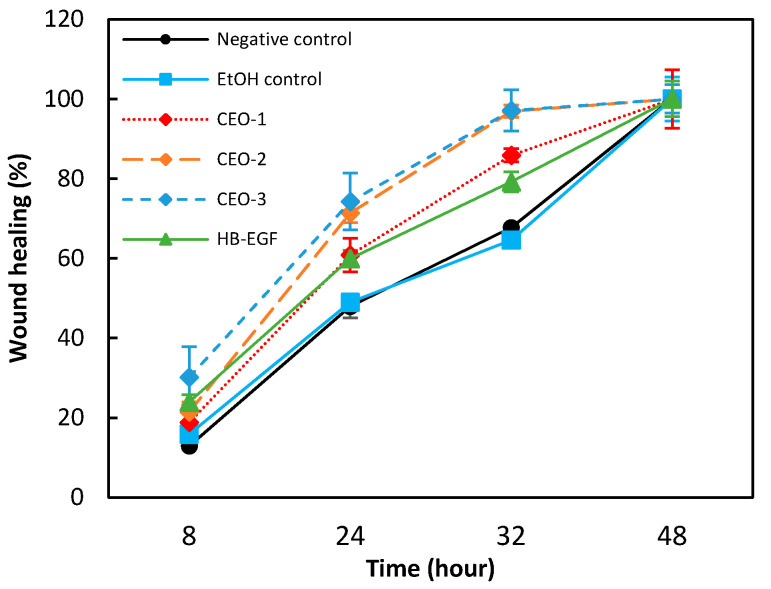
Kinetics of wound healing recorded for the negative and EtOH control groups, the EO (1.8, 4.5 and 9.0 µg/mL) and the HB-EGF positive control.

**Table 1 molecules-27-07966-t001:** Chemical composition of an EO obtained from the oleoresin of *C. schweinfurthii* collected in Gabon (Ntoum region, tree #1).

No.	Compounds	RI Lit. (Non-Polar)	RI Calc. (Non-Polar)	RI Lit. (Polar)^2^	RI Calc. (Polar)	SIM	Content in EO (%)
1	*α*-thujene	930 ^1^	927	nd	nd	93	0.30 ± 0.09
2	*α*-pinene	939 ^1^	935	nd	nd	93	5.87 ± 1.47
3	camphene	954 ^1^	953	nd	nd	93	0.06 ± 0.02
4	sabinene	975 ^1^	974	1124 ± 8	1124	93	7.83 ± 1.32
5	*β*-pinene	979 ^1^	981	1112 ± 7	1118	93	3.77 ± 1.00
6	2-menthene	1004 ± 5 ^2^	1003	nd	nd	93	0.44 ± 0.12
7	*α*-phellandrene	1002 ^1^	1009	1167 ± 9	1168	93	21.88 ± 2.93
8	*δ*^3^-carene	1011 ^1^	1011	1147 ± 7	1150	93	0.39 ± 0.13
9	*α*-terpinene	1017 ^1^	1019	1180 ± 8	1183	93	1.97 ± 0.50
10	*p*-cymene	1024 ^1^	1027	1272 ± 8	1274	117	5.28 ± 0.90
11	limonene	1029 ^1^	1032	1200 ± 7	1203	93	9.09 ± 1.52
12	*β*-phellandrene	1029 ^1^	1034	1211 ± 7	1212	93	5.31 ± 1.44
13	eucalyptol	1031 ^1^	1036	nd	nd	154	0.11 ± 0.06
14	*γ*-terpinene	1059 ^1^	1060	1246 ± 9	1247	93	1.64 ± 0.43
15	4-thujanol	1075 ± 7 ^2^	1073	1469	1461	93	0.59 ± 0.20
16	terpinolene	1088 ^1^	1088	1283 ± 7	1287	93	31.23 ± 4.86
17	*p*-cymenene	1091 ^1^	1092	1444 ± 11	1439	117	0.41 ± 0.13
18	terpinen-4-ol	1177 ^1^	1184	1602 ± 9	1601	93	1.57 ± 0.48
19	*p*-cymen-8-ol	1182 ^1^	1188	1852 ± 13	1847	93	1.37 ± 0.44
20	*α*-terpineol	1188 ^1^	1197	1697 ± 10	1693	93	0.88 ± 0.26

^1^ Determined according to the NIST mass spectral search program for NIST/EPA/NIH EI and NIST tandem mass spectral library—version 2.3, build 4 May 2017. ^2^ Determined according to [23]. nd: not determined.

**Table 2 molecules-27-07966-t002:** Chemical composition of volatiles extracted from the crude oleoresin headspace of *C. schweinfurthii* collected in Gabon (Ntoum region, tree #1).

No.	Compounds	RI Lit. (Non-Polar)	RI Calc. (Non-Polar)	RI Lit. (Polar) ^2^	RI Calc. (Polar)	Content in Crude Oleoresin (%)	Content in EO (%)
1	*α*-thujene	930 ^1^	929	1028 ± 7	1017	0.67 ± 0.30	0.30 ± 0.09
**2**	***α*-pinene**	**939 ^1^**	**936**	**1028 ± 8**	**1013**	**5.68 ± 2.10**	**5.87 ± 1.47**
3	camphene	954 ^1^	953	nd	nd	0.07 ± 0.04	0.06 ± 0.02
**4**	**sabinene**	**975 ^1^**	**975**	**1124 ± 8**	**1115**	**12.44 ± 2.55**	**7.83 ± 1.32**
**5**	***β*-pinene**	**979 ^1^**	**980**	**1112 ± 7**	**1098**	**3.95 ± 0.68**	**3.77 ± 1.00**
6	2-menthene	1004 ± 5 ^2^	1000	nd	nd	0.60 ± 0.10	0.44 ± 0.12
**7**	***α*-phellandrene**	**1002 ^1^**	**1007**	**1167 ± 9**	**1160**	**23.29 ± 1.07**	**21.88 ± 2.93**
8	*δ*^3^-carene	1011 ^1^	1010	1147 ± 7	1143	0.19 ± 0.03	0.39 ± 0.13
9	*α*-terpinene	1017 ^1^	1018	1180 ± 8	1174	1.47 ± 0.06	1.97 ± 0.50
**10**	***p*-cymene**	**1024 ^1^**	**1027**	**1272 ± 8**	**1265**	**7.15 ± 0.66**	**5.28 ± 0.90**
11	**limonene**	**1029 ^1^**	**1032**	**1200 ± 7**	**1191**	**11.72 ± 1.23**	**9.09 ± 1.52**
**12**	***β*-phellandrene**	**1029 ^1^**	**1033**	**1211 ± 7**	**1198**	**2.76 ± 0.27**	**5.31 ± 1.44**
13	eucalyptol	1031 ^1^	1035	nd	nd	0.16 ± 0.07	0.11 ± 0.06
14	*γ*-terpinene	1059 ^1^	1060	1246 ± 9	1239	1.39 ± 0.10	1.64 ± 0.43
15	4-thujanol	1075 ± 7 ^2^	1073	1469	1460	0.11 ± 0.03	0.59 ± 0.20
**16**	**terpinolene**	**1088 ^1^**	**1087**	**1283 ± 7**	**1276**	**27.86 ± 3.27**	**31.23 ± 4.86**
17	*p*-cymenene	1091 ^1^	1091	1444 ± 11	1434	0.24 ± 0.04	0.41 ± 0.13
18	terpinen-4-ol	1177 ^1^	1184	1602 ± 9	1599	0.12 ± 0.02	1.57 ± 0.48
19	*p*-cymen-8-ol	1182 ^1^	1188	1852 ± 13	1845	0.11 ± 0.02	1.37 ± 0.44
20	*α*-terpineol	1188 ^1^	1197	1697 ± 10	1695	0.03 ± 0.01	0.88 ± 0.26

^1^ Determined according to the NIST mass spectral search program for NIST/EPA/NIH EI and NIST tandem mass spectral library—version 2.3, build 4 May 2017. ^2^ Determined according to [23]. nd: not determined.

**Table 3 molecules-27-07966-t003:** CD86 (%) and CD54 (%) values obtained for the in vitro assessment of skin sensitization assay.

	Negative Control	Positive Control	EO					
			C_1_	C_2_	C_3_	C_4_	C_5_	C_6_
CD86 (%)	100	770	212	292	434	494	534	565
CD54 (%)	100	560	316	356	392	472	509	525

**Table 4 molecules-27-07966-t004:** Cytotoxic determination of EO on HaCaT keratinocytes (24 h incubation time, broad EO concentration range).

EO (µg/mL)	Control ^1^	19.53	39.06	78.13	156.25	312.50	625.00	1250.00	2500.00	5000.00
O.D Mean	0.667	0.678	0.666	0.706	0.674	0.532	0.010	0.001	0.001	0.000
St. Dev.	0.034	0.028	0.021	0.028	0.061	0.027	0.005	0.001	0.003	0.000
Viability (%)	100	102	100	106	101	80	1	0	0	0
Cytotoxicity (%)	-	0	0	0	0	20	99	100	100	100

^1^ 10% fetal calf serum medium + 1% EtOH.

**Table 5 molecules-27-07966-t005:** Cytotoxic determination of EO on HaCaT keratinocytes (48 h incubation time, narrow EO concentration range).

EO (µg/mL)	Control ^1^	0.23	0.47	0.94	1.88	3.75	7.50	15.00	30.00	60.00
O.D Mean	0.542	0.523	0.556	0.532	0.525	0.498	0.493	0.418	0.210	0.0000
St. Dev.	0.009	0.009	0.007	0.014	0.010	0.010	0.013	0.015	0.010	0.0000
Viability (%)	100	97	103	98	97	92	91	77	39	0
Cytotoxicity (%)	-	3	0	2	3	8	9	23	61	100

^1^ 10% fetal calf serum medium + 1% EtOH.

**Table 6 molecules-27-07966-t006:** Wound healing determination at 8 h, 24 h, 32 h and 48 h.

	Wound Healing (%)			
	T_1_ (8 h)	T_2_ (24 h)	T_3_ (32 h)	T_4_ (48 h)
Negative control	12.93 ± 0.76	47.92 ± 2.89	67.72 ± 0.63	100.00 ± 1.90
EtOH control	15.87 ± 2.71	48.96 ± 1.29	64.57 ± 1.03	100.00 ± 5.53
C_EO-1_ (1.8 µg/mL)	18.85 ± 4.32	60.84 ± 4.23	85.86 ± 1.65	100.00 ± 7.30
C_EO-2_ (4.5 µg/mL)	21.47 ± 2.51	71.40 ± 2.47	96.95 ± 1.55	100.00 ± 0.76
C_EO-3_ (9.0 µg/mL)	30.18 ± 7.59	74.28 ± 7.14	97.13 ± 5.18	100.00 ± 3.56
HB-EGF (100 ng/mL)	23.96 ± 1.87	59.94 ± 2.11	79.21 ± 2.52	100.00 ± 4.47

**Table 7 molecules-27-07966-t007:** Determination of IL-6 contents.

		IL-6 (pg/mg Prot.)	IL-6 Induced (pg/mg Prot.)	IL-6 Induced (%)	Stat.
(−)UV	Control	63.41 ± 1.59			
(+)UV	Control	1368.55 ± 8.20	1305.14	100	
	C_EO-1_ (1.8 µg/mL)	635.36 ± 6.76	572.22	44	*p* ≤ 0.01
	C_EO-2_ (4.5 µg/mL)	705.68 ± 11.37	642.27	49	*p* ≤ 0.01
	C_EO-3_ (9.0 µg/mL)	700.44 ± 20.16	637.03	49	*p* ≤ 0.01
	Dexamethasone (100 µM)	168.51 ± 5.42	105.10	8	*p* ≤ 0.01

**Table 8 molecules-27-07966-t008:** Determination of TNF-α contents.

		TNF-α (pg/mg Prot.)	TNF-α Induced (pg/mg Prot.)	TNF-α Induced (%)	Stat.
(−)UV	Control	0.57 ± 0.02			
(+)UV	Control	6.07 ± 0.13	5.50	100	
	C_EO-1_ (1.8 µg/mL)	4.07 ± 0.06	3.49	64	*p* ≤ 0.01
	C_EO-2_ (4.5 µg/mL)	4.70 ± 0.14	4.13	75	*p* ≤ 0.01
	C_EO-3_ (9.0 µg/mL)	5.31 ± 0.17	4.74	86	*p* ≤ 0.01
	Dexamethasone (100 µM)	3.17 ± 0.10	2.59	47	*p* ≤ 0.01

## Data Availability

The datasets analyzed in this study are available from the corresponding authors upon reasonable request.

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
