# Peer review of "Wound Healing Potential of an Oleoresin Essential Oil Chemotype from Canarium schweinfurthii Engl."

_molecules, 2022, doi:10.3390/molecules27227966_

Round 1

Reviewer 1 Report

The authors described the wound healing potential of Oleoresin EO chemotype. The article contains very interesting data and merit publication, but before its final shape I will ask for some minor corrections.

1. It would be more attractive if the authors can provide the colorful pictures of graphs.

2. Dear authors kindly add the separate section of discussion, and discussing other papers, not your own results.

3. Can you kindly mention the shortcomings of your this work?

Reviewer 2 Report

The current study investigated the chemical composition of essential oil (EO) extracted from oleoresin of Canarium schweinfurthii. The eye and skin irritancy were evaluated in vitro and in chemico models, and the in vitro modulating potential on a model of wound re-epithelialization was assessed. This EO can reduce the levels of IL-6 and TNF-α, suggesting a possible implication during the inflammation phase of wound healing. The work is of interest and reveals the potential uses of this EO in fragrance and cosmetic fields. The major and minor concerns have been raised for this manuscript, which are listed as follows.

1.      The abstract should be re-written since no conclusion and significance of the work were present.

2.      The spell check is required throughout the manuscript. Eg. “Cytotoxicity on HaCaT 13eratinocytes”on page 13 should beCytotoxicity on HaCaT keratinocytes”

3.      The references need to be formatted according to the journal.

4.      The statistic methods need to be provided.

5.      It is better to show images of wound healing.
